Subject Areas:
biomechanics/behaviour/ecology

Keywords:
sound production, acoustic impulse, amplitude, marine mammal, *Odobenus rosmarus*

Author for correspondence:
Colleen Reichmuth
e-mail: coll@ucsc.edu

# Walruses produce intense impulse sounds by clap-induced cavitation during breeding displays

Ole Næsbye Larsen[1] and Colleen Reichmuth[2]

[1]Department of Biology, University of Southern Denmark, Campusvej 55, DK-5230 Odense M, Denmark
[2]Institute of Marine Sciences, Long Marine Laboratory, 115 McAllister Way, University of California Santa Cruz, Santa Cruz, CA 95060, USA

 ONL, 0000-0002-8325-0982; CR, 0000-0003-0981-6842

Male walruses produce some of the longest continuous reproductive displays known among mammals to convey their physical fitness to potential rivals and possibly to potential mates. Here, we document the ability of a captive walrus to produce intense, rhythmic sounds through a non-vocal pathway involving deliberate, regular collision of the fore flippers. High-speed videography linked to an acoustic onset marker revealed sound production through cavitation, with the acoustic impulse generated by each forceful clap exceeding a peak-to-peak sound level of 200 dB re. 1 µPa. This clapping display is in some ways quite similar to the knocking display more commonly associated with walruses in rut but is produced through a very different mechanism and with much higher amplitudes. While this clapping behaviour has not yet been documented in wild individuals, it has been observed among other mature male walruses living in human care. Production of intense sounds through cavitation has previously been documented only in crustaceans but may also be an effective means of sound production for some aquatic mammals.

## 1. Background

Soundscapes of the oceans are dominated by water motion, weather and distant ship noise and are also influenced by rarer events like lightning, underwater earthquakes, discrete anthropogenic noise sources and the breaking and cracking of sea ice [1–3]. In addition, biological sounds produced by crustaceans, fishes and marine mammals punctuate the soundscapes, which consequently vary considerably with location and time [4]. This means that ambient underwater sound levels can be high over a wide

frequency range. Aquatic animals that communicate using sound therefore must produce signals that are sufficiently powerful to relay their message to intended receivers (e.g. [4]). Aside from social contexts, high-amplitude underwater sounds can also be generated during foraging activity or interaction with predators (e.g. [5]).

Probably the most pervasive and best understood intense transient sounds in the sea are those of some crustaceans, especially snapping shrimp (family Alpheidae) [6]. Due to the low attenuation of sound in water, their impulsive 'snapping' sounds may propagate for kilometres and their near continuous crackling noise may increase the ambient sound level of the sea by up to 20 dB [2,7]. Snapping shrimp use these sounds for communication and the associated water jets for defence and in hunting prey [8,9]. Due to their small size, it has been possible to bring snapping shrimp into the laboratory to investigate their sound-producing mechanism with high-speed video and other sensors (e.g. [10]) and convincingly model the sound production mechanism both physically and mathematically [11–13].

One of a snapping shrimp's frontal claws is disproportionately large and designed such that its moving part (dactyl) can be arrested in an open position while muscular power builds up [14]. When released, the dactyl collides with its immobile counterpart at high speed. The snap produces a fast-moving water jet, which lowers the pressure enough to vaporize water locally to form a bubble that quickly collapses due to the surrounding pressure. The collapse creates an intense sound with a typical spectral frequency peak between 2 and 5 kHz and energy extending to 200 kHz [7]. This process known as *cavitation* takes place in a matter of microseconds [10]. The peak-to-peak source level (SL, the received sound level 1 m from the source) of the impulse sound produced by a single shrimp is extraordinary, up to 190 dB re. 1 µPa [7].

Some marine mammals also produce loud, impulsive sounds under water [15]. So far it has not been possible to document cavitation as a mechanism of sound production in large marine mammals, which cannot as easily be studied in the laboratory. Sound production by cavitation has been suggested for killer whales *Orcinus orca* whose underwater tail slaps into herring schools produce intense 'thud-like' sounds with an average SL of 186 dB re. 1 µPa; such sounds are hypothesized to debilitate fish [16]. While the spectral and temporal characteristics of these tail slaps could possibly be explained by cavitation, no bubbles could be confirmed in the distant underwater video recordings.

Walruses *Odobenus rosmarus* are marine mammals that breed under water near drifting pack ice during the Arctic winter [17]. During the breeding season adult males emit patterned underwater songs that include rhythmic knock sequences punctuated by metallic bell-like sounds. These unusual songs consist of hundreds of short repetitious pulses delivered in repeated stereotyped cycles lasting from several hours to several days at a time [18,19]. These are among the longest and most complex breeding displays known among mammals, but they have rarely been directly observed.

During an ongoing study of the sounds associated with walrus breeding behaviour, we observed an adult male in captivity producing intense transient pulses by clapping his fore flippers together under water, in a similar fashion to human hand clapping [20]. His repeated clapping behaviour occurred coincident with the seasonal production of expected knock sounds of wild walruses and at a similar rate, but the sounds were much louder and produced by an entirely different mechanism. We hypothesized that these impulse sounds were generated by clap-induced cavitation and predicted that sound emission would be associated with the formation of cavitation bubbles, which could be directly observed.

## 2. Methods

During the years 2009–2012, we observed and recorded walrus clapping behaviour. The study animal was a 14–17-year-old mature male Pacific walrus *O. r. divergens*, identified as *Sivuqaq*. He was housed with two to three adult females at Six Flags Discovery Kingdom, a combined zoological and wild rides theme park in Vallejo, California, USA. The park habitat consisted of a large five-sided saltwater pool with adjacent haul-out areas and smaller pools. Water depth in the large pool varied from about 2 m at the walls and sloped to about 4 m in the centre. An observation window, 7.5 m wide and 2.2 m high, spanned one wall about 16 m from the two opposing walls and allowed easy observation of *Sivuqaq's* underwater sound production behaviour. During springtime, he often remained active near (within 1 m) this window stimulated by the presence of visitors or researchers.

We recorded underwater sounds with an ITC 1042 hydrophone with flat frequency response (0.01–100 kHz, ±2.5 dB; sensitivity −201 dB re. 1 V/µPa; International Transducer Corporation Santa

Barbara, CA, USA) placed at 1 m depth within a protective 'well' built into the right side of the pool next to the window to avoid damage from the walruses. Walrus distance from the hydrophone during recording was indicated by black-and-white numbered placards placed along the observation window to mark the horizontal distance from the hydrophone in 1 m intervals. The position of the flippers when clapping was similar in all recordings, approximately 110 cm below the water surface. The hydrophone signal was passed through a broadband Reson VP1000 pre-amplifier (0 dB gain, 1 Hz high-pass; Reson A/S, Slangerup, Denmark) and recorded through the microphone input to a PC computer running Raven Pro 1.3 software (Cornell Lab of Ornithology, Ithaca, NY, USA) at a sampling rate of 96 kHz. Each acoustic file was associated with a video recording obtained with a tripod-mounted conventional video camera (Canon Vixia, type RF H21, 30 frames per second (fps), Melville, NY, USA).

Received peak pressure levels of clap impulses were initially estimated *in situ* by directly measuring the input of the recording chain with a precision sound level meter (type 2250, linear weighting, 48 kHz sampling rate; Brüel & Kjær A/S, Nærum, Denmark). The full bandwidth acoustic recording chain was calibrated following data collection using a reference tone of known received level (RL). Individual pulses were later selected for analysis from underwater recordings based on the review of simultaneous video data, which confirmed the walrus was positioned underwater at the window and clapping the fore flippers together at the moment of each pulse. The recording distance was 2 to 5.5 m from the hydrophone, estimated to the nearest half metre. We inspected each recorded impulsive sound in Raven Pro and measured its received amplitude as a peak–peak level ($L_{p-p}$) relative to a calibration tone in dB re. 1 µPa and noted the recording distance (d) and orientation of the walrus. SL was determined by back-calculating the recorded sound levels at the different distances to 1 m from the walrus using the relation for spherical spreading $SL = L_{p-p} + 20 \log(d)$. This equation applies to the true acoustic far-field. Therefore, this back-calculation will underestimate the actual SL, if the recordings are in the near-field where the amplitude develops with $20 \log(d^2)$. The present recordings probably were in the transition zone between near-field and far-field. We quantified sound pulse parameters in the time and frequency domains as defined in [21] using a custom-written Matlab script (Matlab v. R2020a) and conducted descriptive statistics for acoustic parameters in Prism 7 (GraphPad Software, La Jolla, CA, USA). Acoustic data were visualized in RStudio 1.2.5019 [22].

Clap movements were recorded in fixed-interval 4096 or 8190 ms segments with a high-speed black-and-white motion video camera (Redlake, type MotionPro HS-4; Redlake MASD, LLC, San Diego, CA USA) equipped with a Nikon 28–105 mm AF Nikkor lens (Nikon Corporation, Tokyo, Japan) using Motion Studio software (IDT, Image Acquisition and Processing v. 2.09.02). The recording speed was 1000 fps, corresponding to 1 frame per ms, with 0.9 ms shutter time and wide-open aperture. Video recordings were made at ambient light levels, mainly in the afternoon over a period of 4 days (electronic supplementary material, table S1). To indicate the presence and duration of sound impulses on the high-speed video frames, a custom-built indicator with a microphone and four light emitting diodes (LED) was affixed to the observation window within the field of view of the high-speed camera, and its sensitivity set daily such that the LED display would trigger almost instantaneously (after 0.003 ms) upon receiving intense sound but not in response to typical ambient noise and spoken comments. As the flipper to LED-microphone distance was less than 1 m, the transmission delay was less than 1 ms (electronic supplementary material, table S1). Thus, LED activation in a frame truly marked the acoustic pulse coincident with flipper activity in the same frame. The recorded raw video files were converted to uncompressed AVI files. This high-speed video footage was reviewed manually frame-by-frame using VLC media player (VideoLAN, Open Source Software) for confirmation of cavitation bubble formation, time progression of the cavitation event, and onset and duration of the acoustic pulse (electronic supplementary material, table S1).

# 3. Results

The male walrus *Sivuqaq* performed clapping behaviour by forcefully smacking his fore flippers together in bouts lasting tens of seconds to minutes and with a highly regular inter-clap-interval of 1.2 s (figure 1a and table 1; electronic supplementary material, Video S1). *Sivuqaq* clapped only during the mating season each year from February to May and always under water. His rhythmic clapping behaviour was often associated with sexual arousal (visible erection) and the production of other underwater sounds, including intense knock sounds delivered at a similar rate. The clapping behaviour began at age 13 as *Sivuqaq* approached sexual maturity, but it is unknown whether he spontaneously produced this

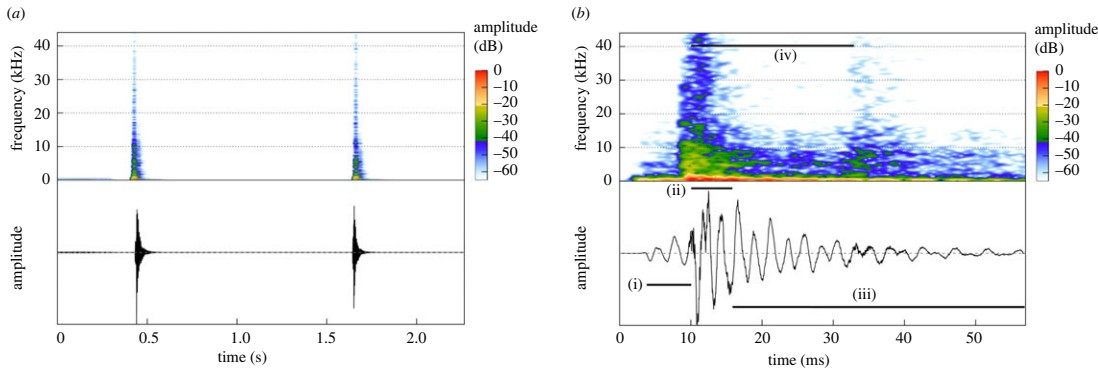

**Figure 1.** Spectrograms (upper) and waveforms (lower) illustrating the acoustic pulse generated by the formation and collapse of the cavitation bubbles produced during the collision of the walrus fore flippers. (*a*) The regularly produced broadband impulses. (*b*) The fine structure of a typical impulse sound showing its three phases and echo from the aquarium back wall. (i) The introductory low-frequency, low-amplitude phase, (ii) the broadband frequency, high-amplitude phase, (iii) the low-frequency damped vibration phase, (iv) the broadband echo from the back walls. The peak-to-peak amplitude of the clap impulse shown in panel (*b*) is 23 751 Pascals (Pa), corresponding to 207 dB re 1 µPa at 1 m.

**Table 1.** Acoustic parameters of 'clap' impulse sounds provided as mean ± s.d., and 5 to 95% data range; $n = 158$ pulses recorded over seven bouts.

| acoustic parameter | mean value ± s.d. | 5–95% range |
|---|---|---|
| received level (dB$_{peak-peak}$ re. 1 µPa at 2–5.5 m) | 191 ± 5.9 | 182–201 |
| source level (dB$_{peak-peak}$ re. 1 µPa at 1 m) | 203 ± 5.2 | 195–211 |
| inter-pulse interval (s) | 1.2 ± 0.1 | 1.1–1.3 |
| 90% energy duration (ms) | 14.9 ± 1.6 | 12.1–17.6 |
| peak frequency (kHz) | 0.5 ± 1.7 | 0.4–0.8 |
| centroid frequency (kHz) | 1.2 ± 0.4 | 0.5–1.9 |
| RMS bandwidth (kHz) | 1.9 ± 0.6 | 0.8–2.8 |

behaviour or if clapping could have been induced or reinforced by visitors to the park. The females he lived with did not clap, and he had never been exposed to conspecific males.

The most spectacular and surprising aspect of his claps was the very intense impulsive sound produced each time his fore flippers collided. These pulses were perceived as sharp and loud even by human observers standing several metres from the 10 cm thick Plexiglass window, behind which the walrus was clapping (electronic supplementary material, Video S1). Measured with a hydrophone in the pool, the distance-dependent RL of 158 recorded claps on average was 191 dB$_{peak-peak}$ re. 1 µPa, corresponding to an average SL of 203 dB$_{peak-peak}$ re. 1 µPa at 1 m distance (table 1; electronic supplementary material, figure S1).

The clap-induced impulse sounds were brief, with 90% of energy (measured from 5% to 95% of accumulated impulse sound energy) emitted within 15 ms (table 1). The pulses were broad-spectrum with sound energy extending to at least 48 kHz (figure 1*a*) but with most energy within a 2 kHz band concentrated at a centroid frequency of 1.2 kHz with an average peak frequency of 500 Hz (table 1).

The fine structure of the single impulse sounds varied but three phases could always be identified (figure 1*b*) even in signals recorded months or years apart. After a 3–6 ms introductory phase (figure 1*b*, (i)) consisting of 2–3 oscillations of 400–500 Hz (zero crossings), there always was a 6–8 ms duration high-amplitude phase with high-frequency oscillations superimposed on the irregular low-frequency oscillations (figure 1*b*, (ii)), after which the signal amplitude decreased relatively regularly as a damped vibration (figure 1*b*, (iii)). About 22 ms after the start of phase two, there was a faint broadband peak in the spectrogram (figure 1*b*, (iv)), which can be attributed to the two-way travel time of the echo propagating at 1500 m s$^{-1}$ from the back walls of the pool about 16 m away.

*Sivuqaq's* near-constant clapping behaviour was interrupted for surfacing to breathe or to swim a few laps from the window to the back walls of the pool (electronic supplementary material, Video S1). The intensity of the behaviour varied from low-amplitude 'lazy' flipper movements just before surfacing to

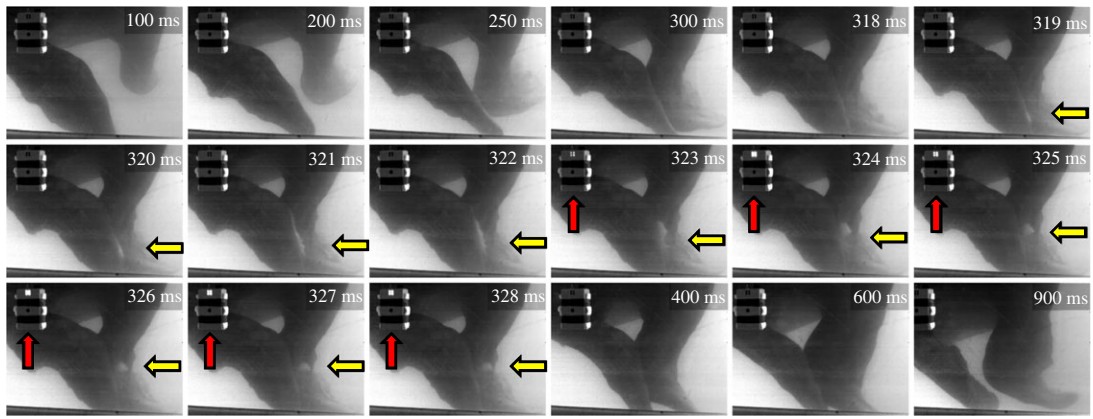

**Figure 2.** Clap-induced cavitation captured by high-speed (1 frame ms$^{-1}$) video. The clapping walrus is turned slightly to the left relative to the high-speed camera placed outside the pool. The right flipper of the walrus (the 'anvil') is shown on the left portion of each image. Horizontal arrows (yellow) show cavitation cloud formation and collapse, vertical arrows (red) show onset and duration of the associated impulse sound. Brightest LED activation is associated with the cavitation cloud 'budding' and shedding. The individual frames are extracted from the same video sequence illustrated in electronic supplementary material, Video S2, which reveals the dynamic progression of the cavitation event. Frame numbers are noted on the image panels.

rather energetic high-amplitude flipper collisions (see movement variation in electronic supplementary material, Video S1). On closer inspection of the clapping behaviour, we observed that it was asymmetrical in the sense that the wrist joint (articulatio radiocarpalis) of the left flipper always hit the palm of the right flipper, such that the left flipper served as 'hammer' and the right flipper as 'anvil'. This orientation of the left flipper blade relative to its trajectory during a clap seemingly minimized the water resistance to the movement and allowed the high-velocity strike.

To further resolve the mechanism of sound production we evaluated high-speed video recordings of single claps (electronic supplementary material, table S1). Recordings of 55 claps were obtained when the colliding walrus flippers remained within the field of view of the camera and in focus. Flipper contact duration during each clapping motion was approximately 185 ms. As these black-and-white recordings were made during daylight hours, the recorded frames often were very dark or very light. However, in all 55 cases, the frames revealed a dynamic bright area forming between the flippers during flipper collision. The 11 most detailed recordings revealed that this bright area was in fact a cloud of smaller structures, probably bubbles, that reflected ambient light better than the surroundings. This area formed upon flipper contact at the distal part of the right flipper and propagated along its edge, where it 'budded out' to be ejected as an almost spherical bubble cloud after reaching the proximal end of the flippers (figure 2 and corresponding electronic supplementary material, Video S2: frames 319 ms to 328 ms), and eventually dissipating after moving away.

The acoustic pulse marked by activation of the LED indicator was captured to high-speed video for 39 of the 55 claps (figure 2, frames 323 ms to 328 ms). While the duration of the LED response was somewhat variable between days based on different trigger settings, the onset of the bright cloud formation always preceded the LED marker of sound generation, with typical latency of $4.8 \pm 1.1$ ms (figure 2; electronic supplementary material, Video S2 and table S1). Most remarkably, the first video frame showing LED activation coincided with the shedding of the bubble cloud in 36 of the 39 recordings (either in the 'budding' or the 'moving away' phase; electronic supplementary material, table S1). From the collective evidence, we interpret the dynamic bright cloud as cavitation bubbles and suggest that the intense impulsive sound was produced by cavitation during the forceful collision of the walrus's flippers.

Associated complete audio and video datasets are available in Dryad [23].

## 4. Discussion

These powerful impulse sounds can be explained by the mechanism of clap-induced cavitation. This is supported by the measured high peak–peak SL of at least 203 dB re. 1 µPa[1] and the presence of spectral

---

[1]This is a conservative estimate; if linear regression of the received levels of the recorded pulses (electronic supplementary material, figure S1a) is used to calculate source level rather than using the far-field calculation, then the average SL estimate is $208 \pm 5.2$ dB$_{\text{peak-peak}}$ re. 1 µPa.

energy extending into the ultrasonic range. High-speed video provides further evidence that each impulse sound is associated with the shedding of a bubble cloud that forms and then dissipates after flipper collision.

Based on our observations, we hypothesize the following sequence of events leading to sound generation. First, flipper collision produces a shock wave in the water that propagates towards the edges of the right flipper (see [20]). Secondly, as the shock wave reaches the flipper's edge, the geometry changes rapidly and the velocity of the shock wave increases enough for the Bernoulli effect to substantially lower the pressure in the water flow. When local pressure declines to a point below the saturated water vapour pressure, microbubbles on the flippers grow to form the observed bright bubble cloud (cavitation inception; [24]) that subsequently moves away and is shed from the flippers. Finally, these bubbles implode due to the higher surrounding pressure of the water, which is approximately 100 kPa (1 atmosphere), corresponding to a maximum possible cavitation sound pressure of 220 dB re. 1 µPa [24].

Inertial cavitation events produced by crustaceans are described on a microsecond scale [10,25], have their peak frequency between 2 and 5 kHz [7], and their peak-to-peak SL in the range 180–190 dB re. 1 µPa [7,26]. By contrast, the walrus clap-induced sound duration is measured on a millisecond scale, the average peak frequency is 500 Hz, and the SLs are at least 203 dB re. 1 µPa. Given these differences, are the measured impulsive clap sounds really created by cavitation?

The resolution and aspect of our video recordings do not allow a detailed analysis of the sound-producing mechanism like in snapping shrimp (e.g. [10,12]). The walrus flipper is about 45 cm long (see electronic supplementary material, Video S1 at 1:29) whereas the length of the snapping shrimp dactyl is only 0.5 cm [13]; the larger size, inertia and drag forces mean that collision contact of the walrus flippers lasts much longer and that the contact area is much larger than in snapping shrimp. In addition, the pressure or shock wave created by flipper collision most likely induces many microbubbles to grow along the flipper edge forming the observed bubble cloud. However, this does not explain the prominent 500 Hz peak frequency (illustrated in figure 1b). Analysis of clapping human hands in air [20] reveals that the precise geometry of the colliding hands, especially whether the hands are domed on impact, determines the peak frequency of the shock wave. So, we hypothesize that the initial 500 Hz wave (figure 1b, (i)) represents the shock wave, which is then boosted by the cavitation event during the second phase of the response (figure 1b, (ii)). After that, the system—left to itself—performs a damped vibration during the third phase (figure 1b, (iii)) that contributes to the full duration sound impulse.

Cavitation forces are known to damage the hammer-like raptorial appendices of mantid shrimp (family Odontodactylidae) used when striking mollusc prey [25]. In these animals, the damage is repaired by moulting. Creating cavitation by forceful clapping for extended periods might similarly damage or alter the walrus flippers. *Sivuqaq* did indeed show shifting lameness and skin eruptions on the right flipper attributed to soft tissue injury over multiple years [27]. Following his death, thorough post-mortem examination did not reveal significant damage to bones or soft tissue, aside from asymmetry as the palm of the right flipper had significant callus formation (V. Hoard, personal communication).

To our knowledge, clapping behaviour has not been reported for wild walruses, raising the question of whether it has a biological function. We suggest that it does, for several reasons. *Sivuqaq's* claps were clearly associated with breeding behaviour as his clapping was not observed outside of rut—the duration and intensity of which was monitored by the testosterone level in his blood [28]. Secondly, several other mature male walruses in captive care have produced variants of this clapping behaviour during the breeding season while performing stereotyped acoustic displays.[2] In addition, like *Sivuqaq*, mature male walruses in the wild and in captivity produce intense impulse sounds known as knocks in highly stereotypical sequences or songs (e.g. [18,29,30]). Knocks are low-frequency pulses (most energy between 500 and 2000 Hz, but extending to at least 24 kHz), often repeated in rhythmic sequences at rates of 0.8–1.2 Hz with SL of 184–186 dB$_{peak-peak}$ re. 1 µPa [31,32]. *Sivuqaq* produced both intense knocks and clap sounds, often at the same time. We therefore hypothesize that walrus clapping behaviour also occurs in the wild, is intrinsically motivated (since *Sivuqaq* performed without ever having observed adult males) and may have a biological function.

Walruses tend to perform their reproductive behaviour in the moving pack ice of the high Arctic from January to March [17,19] when light levels are low and underwater noise levels are high (e.g. [33]). Such habitat is challenging to access and confirmed observations and sound recordings of walrus reproductive

[2]Personal communications concerning five male walruses from zoological staff: L. Triggs, Point Defiance Zoo and Aquarium, 1 May 2011; M. Synnott, Sea World San Diego, 16 August 2019; W. Winhall, SeaWorld San Diego, 16 August 2019; M. Shoemaker, SeaWorld Orlando, 29 October 2020.

displays during rut are scarce [17,18,29,30,34]. However, a regularly repeated, powerful sound impulse seems ideal for advertising the presence of a competitive male walrus through the irregular acoustic background. We hypothesize that the surprising regularity of *Sivuqaq's* clap pulse production could result from water resistance (drag) to forelimb movement constraining the maximum clapping rate or could be entrained to the similar heart rate during submersion [35]. Clap regularity and intensity during reproductive displays could signal male fitness to deter competitors and possibly attract potential mates within a mating system based on female-defence polygyny [17,30] and may substitute for or enhance knocking displays during intrasexual competition.

Intense impulsive sounds are produced by several marine mammals during reproductive behaviour. Such sounds can be produced without specialized sound production structures. Male grey seals *Halichoerus grypus* have recently been filmed producing similar sharp fore flipper claps in the wild, where the clap sounds seemingly are used for communication [36]; their sound production mechanism has not been investigated but the clap sounds could be produced by cavitation. Harbour seal males *Phoca vitulina* slap the water's surface during reproductive displays, creating an attention-getting 'fire-cracker' sound with underwater SLs of 186–199 dB$_{peak-peak}$ re. 1 µPa [37,38]. Humpback whales *Megaptera novaeangliae* [39,40], North Atlantic right whales *Eubalaena glacialis* [41] and killer whales *Orcinus orca* [42] also create intense impulsive sounds by pectoral slapping of the water surface but this sound production has not been studied in detail. Thus, it seems possible that other marine mammals can produce intense impulsive sounds through cavitation.

Ethics. Walruses at Six Flags Discovery Kingdom were displayed under authorization from the United States Fish and Wildlife Service. Audio, video, observational and veterinary data were obtained without harm to animals, with oversight and approval from the Animal Welfare Committee at the park and the Institutional Animal Care and Use Committee at the University of California Santa Cruz.

Data accessibility. Original sound and high-speed video recordings, Matlab code for acoustic analyses, and summary spreadsheets are published in the Dryad repository: Reichmuth, Colleen; Larsen, Ole Næsbye (2020), Walruses produce intense impulse sounds by clap-induced cavitation during breeding displays, Dryad, Dataset, https://doi.org/10.7291/D1VT11 [23].

Authors' contributions. O.N.L. and C.R. contributed to the design and implementation of the research, to the analysis of the results and to the writing of the manuscript.

Competing interests. We declare we have no competing interests.

Funding. This work was supported by the Danish Council for Independent Research | Natural Sciences (grant no. 10-093073 to O.N.L.).

Acknowledgements. The authors thank Michael Muraco, Director of Animal Care at Six Flags Discovery Kingdom in Vallejo, CA for providing access to the walruses and Leah Coombs and the animal care staff for valuable assistance in the park. William Hughes and Jason Mulsow assisted us significantly with data collection. We are especially grateful to D. Felipe Gaitan and to Peter Møller Juhl for helping us interpret the high-speed video recordings of cavitation events and to Frank Mortensen for building the LED-sound indicator. Jeppe Have Rasmussen, Magnus Wahlberg and Jillian Sills assisted us with video conversion and acoustic analysis, and Magnus Wahlberg, Caroline Casey and Edward Miller provided helpful comments on the manuscript. Two anonymous reviewers provided comments that improved this manuscript.

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
