## [Peer Review File · Royal Society Open Science]

Review History

RSOS-210197.R0 (Original submission)

Review form: Reviewer 1 (Christopher Clark)

Is the manuscript scientifically sound in its present form?

Yes

Are the interpretations and conclusions justified by the results?

Yes

Is the language acceptable?

Yes

Do you have any ethical concerns with this paper?

No

Have you any concerns about statistical analyses in this paper?

No

Recommendation?

Major revision is needed (please make suggestions in comments)

Comments to the Author(s)

Dear Editor,

The authors have addressed most of my comments/criticisms from the previous round of review. There remain two closely-related issues that the authors have not addressed. One is a somewhat important technical error which the authors absolutely must fix (e.g. in Fig 2). The other is related, which is that the authors haven't really analyzed their videos very much. Why they report recording 55 high-speed videos is not clear to me, if they weren't going to analyze them. In the response to review they state that they have uploaded their videos to an online repository, but uploading un-analyzed data is an inferior approach to actually analyzing their videos for insight they yield into the process described in the paper, and then presenting it concisely in their paper. They have basically analyzed 1 video, the one shown in Figure 2 and in the supplemental. When I watched it, there was clearly a bubble that formed, and it had to be a cavitation bubble; it convinced me of the basic correctness of the main result of this paper. The reason to record and report on 55 videos (rather than 1 or a few) is to analyze variation. But anyway, let me come back to this below. First the glaring technical error:

1) On line 141 the authors mistakenly use the word "instantaneous", implying that the speed of sound is nearly infinite. It is not. I know they know this, because elsewhere they make correct statements about acoustic reflections within Sivuqaq's pool. When you synchronize video with sound, you absolutely must address the finite speed of sound (versus the nearly infinite speed of light).

Clark 2016 expressly addresses how to synchronize high-speed video with sound recordings: he gives the equation $l = nd / c$ for adjusting time delay to synchronize high speed video with sound on page 89. l is the lag in video frames, n is the framerate of the high speed video camera (1000 fps here), d is the distance between the source and the microphone (2 to 5.5 m, according to the text of the paper), and c is the speed of sound (1500 m/s in water). Put these numbers into the equation and you get 3.67 frames (when Sivuqaq is 5.5 m from the mic; less when he's closer).

Clark 2016: https://link.springer.com/chapter/10.1007/978-3-319-27721-9_4

A corollary: Sivuqaq has to be within 0.75 m of the microphone for sound travel from his flippers to the microphone in less than less than half a frame of the high speed video, and thus be functionally instantaneous relative to the recording rate of 1000 fps.

The results and Figure 2 need to be modified once the authors address this issue. If it's really the cavitation cloud that produces the sound, there should NOT be a 4 ms delay between onset of one and onset of the other! If you've correctly identified the mechanism in the video that produces the sound, the physical event in the video will be simultaneous with the sound (plus or minus your measurement error, which for video will depend mainly on framerate and a bit on shutter speed)

(2) the authors have not analyzed the high speed videos very much. In their response to review they report that all of the videos were highly similar. I am a bit skeptical of this claim. When I have analyzed similar datasets, what I have done is matched up variation in the video with variation in the sound (after correcting for the time delay between video and sound, per the above major criticism). This variation might be slight, but is it really zero? If the events shown in the video really are the source of the sounds recorded by the mic, there should be an exact

correspondence between video events and sounds. But they first have to first deal with issue #1 before they could look at fine-scale timing of the sounds vs the videos they go with, since variation in the walrus-to-microphone distance will obscure more fine-scale timing variables. Also the authors say they can't measure flipper velocity from the videos due to lack of an appropriate scale. Fair enough, but the most basic way to analyze videos like these are to count frames of events, such as the (apparent) timing of flipper-flipper contact, and the appearance of the cavitation bubbles. If these events really are invariant when filmed at 1000 fps, a supplemental table with some numbers (from sound recordings and from video) would validate and make much stronger the authors' assertion.

Ultimately this issue, that they have recorded 55 videos but present almost nothing about them, is not a fatal one to this paper. It took exactly 1 video to convince me that this sound is produced by cavitation. But it is strange to me that the authors would record 55 videos, synch them (or a significant subset of them) to the sound, but then not present some simple analyses of what their synched videos plus sounds show.

Minor comments

Line 52: They don't actually mean acoustic power, do they? I suggest using the word 'amplitude' rather than power, since the relevant physical parameter is pressure (this is what vertebrate ears detect), not the actual power.

Line 80: rephrase as "bell-like" rather than bell, there was not literally a bell

Line 87: for some reason this read funny to me. Consider rephrasing as " expected knock sounds of wild walruses, and at a...."

Apologies I didn't notice this was missing in the previous review: what was the shutterspeed of the high speed video? Since you were in a dark environment, I assume shutterspeed was also 1/1000 sec? And presumably the aperture was wide open?

Paragraph 166-173: This paragraph offers a detailed description of the sound Sivuqaq produced. What is still missing is a comparison to the high speed videos and timing of events in the HSV. In their response to review the authors explained why quantifying the flipper velocity would be uncertain, but the later description of what the videos shows is mostly qualitative rather than quantitative analyses of the videos on lines 184-190. "the most detailed recordings" -> replace this with an N = ? of how many were the most detailed.

Line 195: but this finding of 4.8 +/- 1.1 ms is with the mistaken assertion that the sound traveled instantaneously from the walrus to the detector. Accounting for the speed of sound in water (per the above) will reduce this latency, I assume to within your measurement error (of +/- 1/2 frame, or 0.5 ms). After all, if this is really a causal relationship the authors have found, there should be *no* latency. If the authors have the walrus-microphone distance for each individual clap, they could calculate the latencies for individual claps. if they don't, then the best they can do is to calculate an average latency.

Line 205: dissolve is the wrong word. Dissipate?

Review form: Reviewer 2

Is the manuscript scientifically sound in its present form?

Yes

Are the interpretations and conclusions justified by the results?

Yes

Is the language acceptable?

Yes

Do you have any ethical concerns with this paper?

No

Have you any concerns about statistical analyses in this paper?

No

Recommendation?

Accept with minor revision (please list in comments)

Comments to the Author(s)

This manuscript is greatly improved from the first version I reviewed. In general, the authors' revisions have adequately addressed my concerns from my previous review.

One exception -

Methods: this section is missing information such as the total number of recordings taken, how long the recordings were, how often they were taken, etc.

Response: These data are all provided in Dryad. See reference on new line 278-281.

I don't think this is acceptable. This information is about basic methods, not results. The readers should be able to see an overview of your methods in the paper itself without having to download your 13.7 GB Dryad dataset. It took over an hour to download - 45 minutes waiting for "Dryad is now preparing your download", then another 16-17 minutes for the actual download. And I have a reasonably fast internet connection. This information could be provided in a sentence or two.

Decision letter (RSOS-210197.R0)

Dear Dr Reichmuth

On behalf of the Editors, we are pleased to inform you that your Manuscript RSOS-210197 "Walrus produce intense impulse sounds by clap-induced cavitation during breeding displays" has been accepted for publication in Royal Society Open Science subject to minor revision in accordance with the referees' reports. Please find the referees' comments along with any feedback from the Editors below my signature.

Please submit your revised manuscript and required files (see below) no later than 7 days from today's (ie 19-Apr-2021) date. Note: the ScholarOne system will 'lock' if submission of the revision is attempted 7 or more days after the deadline. If you do not think you will be able to meet this deadline please contact the editorial office immediately.

on behalf of Dr Ari Friedlaender (Associate Editor) and Kevin Padian (Subject Editor)
openscience@royalsociety.org

Associate Editor Comments to Author (Dr Ari Friedlaender):

Both reviewers lauded the efforts to incorporate previous comments, and the submission is greater enhanced from that work. However, there are still several methodological edits that require attention as they are fundamental to being able to interpret or reproduce the study and I would like these to be addressed. As well, one reviewer has a comment/suggestion that would require major revision and I would like to hear from the authors whether they believe this is of value to the current submission and if not, why they would choose not to conduct the further analyses requested.

Aside from these, I believe the submission to be well-written, scientifically sound, and of interest.

Thank you.
Ari S. Friedlaender

Reviewer comments to Author:
Reviewer: 1
Comments to the Author(s)

Dear Editor,

The authors have addressed most of my comments/criticisms from the previous round of review. There remain two closely-related issues that the authors have not addressed. One is a somewhat important technical error which the authors absolutely must fix (e.g. in Fig 2). The other is related, which is that the authors haven't really analyzed their videos very much. Why they report recording 55 high-speed videos is not clear to me, if they weren't going to analyze them. In the response to review they state that they have uploaded their videos to an online repository, but uploading un-analyzed data is an inferior approach to actually analyzing their videos for insight they yield into the process described in the paper, and then presenting it concisely in their

paper. They have basically analyzed 1 video, the one shown in Figure 2 and in the supplemental. When I watched it, there was clearly a bubble that formed, and it had to be a cavitation bubble; it convinced me of the basic correctness of the main result of this paper. The reason to record and report on 55 videos (rather than 1 or a few) is to analyze variation. But anyway, let me come back to this below. First the glaring technical error:

1) On line 141 the authors mistakenly use the word "instantaneous", implying that the speed of sound is nearly infinite. It is not. I know they know this, because elsewhere they make correct statements about acoustic reflections within Sivuaq's pool. When you synchronize video with sound, you absolutely must address the finite speed of sound (versus the nearly infinite speed of light).

Clark 2016 expressly addresses how to synchronize high-speed video with sound recordings: he gives the equation $l = nd / c$ for adjusting time delay to synchronize high speed video with sound on page 89. l is the lag in video frames, n is the framerate of the high speed video camera (1000 fps here), d is the distance between the source and the microphone (2 to 5.5 m, according to the text of the paper), and c is the speed of sound (1500 m/s in water). Put these numbers into the equation and you get 3.67 frames (when Sivuaq is 5.5 m from the mic; less when he's closer).

Clark 2016: https://link.springer.com/chapter/10.1007/978-3-319-27721-9_4

A corollary: Sivuaq has to be within 0.75 m of the microphone for sound travel from his flippers to the microphone in less than less than half a frame of the high speed video, and thus be functionally instantaneous relative to the recording rate of 1000 fps.

The results and Figure 2 need to be modified once the authors address this issue. If it's really the cavitation cloud that produces the sound, there should NOT be a 4 ms delay between onset of one and onset of the other! If you've correctly identified the mechanism in the video that produces the sound, the physical event in the video will be simultaneous with the sound (plus or minus your measurement error, which for video will depend mainly on framerate and a bit on shutter speed)

(2) the authors have not analyzed the high speed videos very much. In their response to review they report that all of the videos were highly similar. I am a bit skeptical of this claim. When I have analyzed similar datasets, what I have done is matched up variation in the video with variation in the sound (after correcting for the time delay between video and sound, per the above major criticism). This variation might be slight, but is it really zero? If the events shown in the video really are the source of the sounds recorded by the mic, there should be an exact correspondence between video events and sounds. But they first have to first deal with issue #1 before they could look at fine-scale timing of the sounds vs the videos they go with, since variation in the walrus-to-microphone distance will obscure more fine-scale timing variables.

Also the authors say they can't measure flipper velocity from the videos due to lack of an appropriate scale. Fair enough, but the most basic way to analyze videos like these are to count frames of events, such as the (apparent) timing of flipper-flipper contact, and the appearance of the cavitation bubbles. If these events really are invariant when filmed at 1000 fps, a supplemental table with some numbers (from sound recordings and from video) would validate and make much stronger the authors' assertion.

Ultimately this issue, that they have recorded 55 videos but present almost nothing about them, is not a fatal one to this paper. It took exactly 1 video to convince me that this sound is produced by cavitation. But it is strange to me that the authors would record 55 videos, synch them (or a significant subset of them) to the sound, but then not present some simple analyses of what their synched videos plus sounds show.

Minor comments

Line 52: They don't actually mean acoustic power, do they? I suggest using the word 'amplitude' rather than power, since the relevant physical parameter is pressure (this is what vertebrate ears detect), not the actual power.

Line 80: rephrase as "bell-like" rather than bell, there was not literally a bell

Line 87: for some reason this read funny to me. Consider rephrasing as "expected knock sounds of wild walruses, and at a...."

Apologies I didn't notice this was missing in the previous review: what was the shutter speed of the high speed video? Since you were in a dark environment, I assume shutter speed was also 1/1000 sec? And presumably the aperture was wide open?

Paragraph 166-173: This paragraph offers a detailed description of the sound Sivuqaq produced. What is still missing is a comparison to the high speed videos and timing of events in the HSV. In their response to review the authors explained why quantifying the flipper velocity would be uncertain, but the later description of what the videos shows is mostly qualitative rather than quantitative analyses of the videos on lines 184-190. "the most detailed recordings" -> replace this with an N = ? of how many were the most detailed.

Line 195: but this finding of 4.8 +/- 1.1 ms is with the mistaken assertion that the sound traveled instantaneously from the walrus to the detector. Accounting for the speed of sound in water (per the above) will reduce this latency, I assume to within your measurement error (of +/- 1/2 frame, or 0.5 ms). After all, if this is really a causal relationship the authors have found, there should be *no* latency. If the authors have the walrus-microphone distance for each individual clap, they could calculate the latencies for individual claps. If they don't, then the best they can do is to calculate an average latency.

Line 205: dissolve is the wrong word. Dissipate?

Reviewer: 2

Comments to the Author(s)

This manuscript is greatly improved from the first version I reviewed. In general, the authors' revisions have adequately addressed my concerns from my previous review.

One exception -

Methods: this section is missing information such as the total number of recordings taken, how long the recordings were, how often they were taken, etc.

Response: These data are all provided in Dryad. See reference on new line 278-281.

I don't think this is acceptable. This information is about basic methods, not results. The readers should be able to see an overview of your methods in the paper itself without having to download your 13.7 GB Dryad dataset. It took over an hour to download - 45 minutes waiting for "Dryad is now preparing your download", then another 16-17 minutes for the actual download. And I have a reasonably fast internet connection. This information could be provided in a sentence or two.

===PREPARING YOUR MANUSCRIPT===

===PREPARING YOUR REVISION IN SCHOLARONE===

<https://royalsociety.org/journals/authors/author-guidelines/#supplementary-material> to include a suitable title and informative caption. An example of appropriate titling and captioning may be found at https://figshare.com/articles/Table_S2_from_Is_there_a_trade-off_between_peak_performance_and_performance_breadth_across_temperatures_for_aerobic_scooping_in_teleost_fishes_/3843624.

Author's Response to Decision Letter for (RSOS-210197.R0)

See Appendix A.

Decision letter (RSOS-210197.R1)

Dear Dr Reichmuth,

It is a pleasure to accept your manuscript entitled "Walrus produce intense impulse sounds by clap-induced cavitation during breeding displays" in its current form for publication in Royal Society Open Science. The comments of the reviewer(s) who reviewed your manuscript are included at the foot of this letter.

You can expect to receive a proof of your article in the near future. Please contact the editorial office (openscience@royalsociety.org) and the production office (openscience_proofs@royalsociety.org) to let us know if you are likely to be away from e-mail contact – if you are going to be away, please nominate a co-author (if available) to manage the proofing process, and ensure they are copied into your email to the journal.

on behalf of Dr Ari Friedlaender (Associate Editor) and Kevin Padian (Subject Editor)
openscience@royalsociety.org

Editor comments:
Thanks for your revisions and congratulations!

Associate Editor Comments to Author (Dr Ari Friedlaender):

Associate Editor

Comments to the Author:

To the Authors,

I appreciate the care and diligence you have taken to address all reviewer comments and suggestions. I am confident that the revised version on the study not only incorporates these edits, but that they have added to the overall clarity and strength of the presentation of information. Based on this, I am happy to recommend publication of your work.

Congratulations on a fine study that will be of interest to a broad range of readers of RSOS.

Thank you for your efforts.

Ari S. Friedlaender

Appendix A

JOSEPH M. LONG MARINE LABORATORY
INSTITUTE OF MARINE SCIENCES
UNIVERSITY OF CALIFORNIA, SANTA CRUZ
115 McALLISTER WAY | SANTA CRUZ, CALIFORNIA 95060
831/459-2883 | 831/459-3383 FAX

26 April 2021

Dear Dr Friedlaender,

Thank you for your response to our revised manuscript. We appreciate the opportunity to resolve these issues quickly and move this manuscript forward to publication. We also appreciate the thoughtful criticisms of the reviewers which, as you have noted, have improved the quality of our work. Please see our responses below (in blue) to the reviewer comments.

We note that the major comment provided by the first reviewer seems to be a misunderstanding that we have clarified below. In addition, to make this information more accessible, we have elected to include a supplementary table (Table S1) with the original video analyses of all recorded claps and the requested further analyses. While much of this information is available at the Dryad reference provided, we agree that it will be simpler for a reader to find this table linked to the RSOS manuscript.

We have uploaded the revised manuscript and trust that you will find it suitable for rapid publication in RSOS.

Sincerely,
Ole Næsbye Larsen and Colleen Reichmuth

Associate Editor Comments to Author (Dr Ari Friedlaender):

Both reviewers lauded the efforts to incorporate previous comments, and the submission is greater enhanced from that work. However, there are still several methodological edits that require attention as they are fundamental to being able to interpret or reproduce the study and I would like these to be addressed. As well, one reviewer has a comment/suggestion that would require major revision and I would like to hear from the authors whether they believe this is of value to the current submission and if not, why they would choose not to conduct the further analyses requested.

Aside from these, I believe the submission to be well-written, scientifically sound, and of interest.

Thank you.
Ari S. Friedlaender

Reviewer comments to Author:

Reviewer: 1

Comments to the Author(s)

Dear Editor,

The authors have addressed most of my comments/criticisms from the previous round of review. There remain two closely-related issues that the authors have not addressed. One is a somewhat important technical error which the authors absolutely must fix (e.g. in Fig 2). The other is related, which is that the authors haven't really analyzed their videos very much. Why they report recording 55 high-speed videos is not clear to me, if they weren't going to analyze them. In the response to review they state that they have uploaded their videos to an online repository, but uploading un-analyzed data is an inferior approach to actually analyzing their videos for insight they yield into the process described in the paper, and then presenting it concisely in their paper. They have basically analyzed 1 video, the one shown in Figure 2 and in the supplemental. When I watched it, there was clearly a bubble that formed, and it had to be a cavitation bubble; it convinced me of the basic correctness of the main result of this paper. The reason to record and report on 55 videos (rather than 1 or a few) is to analyze variation. But anyway, let me come back to this below. First the glaring technical error:

Answer (also to Reviewer #2): We were not aware that it was so cumbersome to download material from Dryad. We therefore have included a table (Table S1) in the Supplementary Electronic Material with data on high-speed video recordings and subsequent analysis, to which we refer below and in the manuscript Results.

(1) On line 141 the authors mistakenly use the word "instantaneous", implying that the speed of sound is nearly infinite. It is not. I know they know this, because elsewhere they make correct statements about acoustic reflections within Sivuqaq's pool. When you synchronize video with sound, you absolutely must address the finite speed of sound (versus the nearly infinite speed of light).

Answer: The reviewer seems to have misunderstood the text we provided. The word "instantaneous" used here refers to the processing time of the microphone-LED indicator. When sound arrives at the microphone, it takes 0.003 ms until the LEDs light up. '... its sensitivity set such that the LED display would trigger instantaneously (within 0.003 ms) upon receiving intense sound (light activation above a high acoustic threshold) but not in response to typical noise and spoken comments.' We thought that this formulation was sufficiently clear. The high setting (low sensitivity) on the microphone-LED indicator might cause a delay in activation until the sound pressure associated with cavitation reached the acoustic threshold of the LED light emitter. To avoid any confusion, we have changed the text to '... its sensitivity set such that the LED display would trigger almost instantaneously (after 0.003 ms) upon receiving intense sound but not in response to typical ambient noise and spoken comments'.

Clark 2016 (https://link.springer.com/chapter/10.1007/978-3-319-27721-9_4) **expressly addresses** how to synchronize high-speed video with sound recordings: he gives the equation $l = nd / c$ for adjusting time delay to synchronize high speed video with sound on page 89. l is the lag in video frames, n is the framerate of the high speed video camera (1000 fps here), d is the distance between the source and the microphone (2 to 5.5 m, according to the text of the paper),

and c is the speed of sound (1500 m/s in water). Put these numbers into the equation and you get 3.67 frames (when Sivuqaq is 5.5 m from the mic; less when he's closer).

Answer: There is some confusion by the reviewer about the placement of the hydrophone used to record sounds, and the microphone used to activate the light emitter. The distances of 2-5.5 m refer to the horizontal distances between colliding flippers and the hydrophone in the well, when recording the sound levels at different distances for back-calculating source levels. In the high-speed video recordings, we used the microphone-LED indicator placed on the window to show the presence of sound on the video frames. For the flippers and the microphone-LED indicator to be present simultaneously in the picture (field of view), their mutual x-y distance in the window plane ranged from 20 to 60 cm (using the dimensions of the microphone-LED indicator, 10 cm by 7 cm, as a yardstick). The flipper distance perpendicular to the window plane was more uncertain but did not exceed 1 m, as Sivuqaq often rested his head on the window while clapping (see Video S1; Table S1, column Q). To get an estimate of the distance-caused delay between flipper position and LED-microphone we calculated the distances for all individual claps if the flippers were 0.5 m and 1.0 m away from the window. The mean distances were 61 cm and 106 cm, which at a sound propagation velocity of $c = 1,500$ m/s corresponds to delays of 0.4 and 0.7 ms, respectively (see Table S1, column L).

To clarify the issue, we inserted in lines 143-145 *'As the flipper to LED-microphone distance was less than 1 m, the transmission delay was less than 1 ms (Table S1, L). So, LED activation in a frame truly marked the the acoustic pulse coincident with flipper activity in the same frame.'*

Please see further discussion of the 0.4 ms delay in the answer below.

A corollary: Sivuqaq has to be within 0.75 m of the microphone for sound travel from his flippers to the microphone in less than less than half a frame of the high speed video, and thus be functionally instantaneous relative to the recording rate of 1000 fps.

Answer: Exactly, this is what we find – see answer above.

The results and Figure 2 need to be modified once the authors address this issue. If it's really the cavitation cloud that produces the sound, there should NOT be a 4 ms delay between onset of one and onset of the other! If you've correctly identified the mechanism in the video that produces the sound, the physical event in the video will be simultaneous with the sound (plus or minus your measurement error, which for video will depend mainly on framerate and a bit on shutter speed)

Answer: Since the propagation time from flippers to Microphone-LED and the response time of the microphone-LED together is less than 1 ms, this cannot explain the average 4 ms delay (see Table S1, column N). However, the sensitivity of the LED-microphone was set each day depending on ambient noise (noisy visitors). A low sensitivity could mean that the LED-microphone would not respond to the initial low amplitude part of the acoustic pulse, see phase "i" in Fig. 1b. This could explain at least part of the 4 ms delay. Another question is when the cavitation bubbles actually implode to create the high amplitude sounds. Do they implode when the cloud is first observed propagating along the edge of the flipper or only later? Here it is interesting to note that the first LED activation occurs, when the cavitation cloud starts to 'bud' and move away from the flippers, which happens in 36 out of 39 recordings (see Table S1, column O) and the absolutely highest LED light intensity occurs when the cloud starts to dissipate when implosion certainly must have occurred. Here, the sensitivity setting of the microphone-LED does not interfere. We therefore do not see any reason for changing Fig. 2. However, we emphasized in the caption to Fig. 2 *'Highest LED activation is associated with cavitation cloud 'budding' and shedding.'*

We suggest that our hypothesis of a two-step mechanistic process (proposed in lines 217-227) is entirely consistent with the reported acoustic and video observations.

(2) The authors have not analyzed the high-speed videos very much. In their response to review they report that all of the videos were highly similar. I am a bit skeptical of this claim. When I have analyzed similar datasets, what I have done is matched up variation in the video with variation in the sound (after correcting for the time delay between video and sound, per the above major criticism). This variation might be slight, but is it really zero? If the events shown in the video really are the source of the sounds recorded by the mic, there should be an exact correspondence between video events and sounds. But they first have to first deal with issue #1 before they could look at fine-scale timing of the sounds vs the videos they go with, since variation in the walrus-to-microphone distance will obscure more fine-scale timing variables. Also, the authors say they can't measure flipper velocity from the videos due to lack of an appropriate scale. Fair enough, but the most basic way to analyze videos like these are to count frames of events, such as the (apparent) timing of flipper-flipper contact, and the appearance of the cavitation bubbles. If these events really are invariant when filmed at 1000 fps, a supplemental table with some numbers (from sound recordings and from video) would validate and make much stronger the authors' assertion.

Ultimately this issue, that they have recorded 55 videos but present almost nothing about them, is not a fatal one to this paper. It took exactly 1 video to convince me that this sound is produced by cavitation. But it is strange to me that the authors would record 55 videos, synch them (or a significant subset of them) to the sound, but then not present some simple analyses of what their synched videos plus sounds show.

Answer: We appreciate that this reviewer is convinced about the mechanism of sound production, which is our primary intention with this manuscript. As noted, we did evaluate every one of the 55 videos, which are now published and publicly available as a DRYAD data set including an analysis table. However, this table does not include the suggested details of timing that the reviewer offers above.

Based on the reviewer feedback, we have elected to add that information, where available, to our video analysis table and to publish this data table as an electronic supplement to this RSOS article (Table S1). We emphasize again that the video sample included in this manuscript is not an outlier but rather a representation of the typical sequence of events we observed in our data. Reference to this table also provides methodological data requested by Reviewer 2 (below). Thus, we have added the following columns of data to Table S1 and included data in the Results where applicable:

- 1) The estimated clap duration (flipper contact duration by number of frames), which is very uncertain as it is extremely difficult to define time of flipper separation (column K)
- 2) The estimated distance from the contact point of the flippers to the light indicator placed on the window and the average delay for two flipper distances (0.5 and 1.0 m) from the window (column L)
- 3) The number of frames (ms) from the frame with first emergence of bubbles propagating along the edge of the flipper to the frame with first LED activation (column N)
- 4) The phase in cloud development when LED activation first occurs (column O). The phases are: (1) CB moves up towards edge; (2) CB concentrates (in a jet?); (3) The bubble cloud starts the "bud" out from the flippers; (4) The "bud" forms an almost spherical cloud that starts to move away from the flippers; (5) The cloud starts to dissipate
- 5) The number of frames that the LED indicator is activated, indicating the duration of the sound pulse that exceeds the LED-microphone threshold setting (column P).
- 6) A 'rating' of each video file is now associated with the comments for each event. This indicates that there were 11 exemplars of the highest quality in our data record, similar to that shown in Fig. 2 (column Q).

This information, added to the original data in this table (Table S1) and now made available as a supplement to this article, should provide any reader with the ability to confirm the illustrative data provided in the main section of this short manuscript.

We have included this information in the Results section, the last paragraph of which now reads: *“The acoustic pulse marked by activation of the LED indicator was captured to high-speed video for 39 of the 55 claps (see Fig. 2, frames 323 ms to 328 ms). While the duration of the LED response was somewhat variable between days based on different trigger settings, the onset of the bright cloud formation always preceded the LED marker of sound generation, with typical latency of 4.8 ± 1.1 ms (Fig. 2; Video S2; Table S1). Most remarkably, the first video frame showing LED activation coincided with shedding of the bubble cloud in 36 of the 39 recordings (either in the “budding” or the “moving away” phase; Table S1). From the collective evidence, we interpret the dynamic bright cloud as cavitation bubbles and suggest that the intense impulsive sound was produced by cavitation during the forceful collision of the walrus’s flippers.”*

Minor comments

Line 52: They don't actually mean acoustic power, do they? I suggest using the word 'amplitude' rather than power, since the relevant physical parameter is pressure (this is what vertebrate ears detect), not the actual power.

Answer: We we agree with the reviewer that we are referring to powerful amplitude here, however, this is clear from the previous and following sentences, and we have elected not to change the wording on this line.

Line 80: rephrase as "bell-like" rather than bell, there was not literally a bell

Answer: We have altered the text on Line 80: *'... punctuated by metallic bell-like sounds ...'*

Line 87: for some reason this read funny to me. Consider rephrasing as "expected knock sounds of wild walruses, and at a...."

Answer: We have altered the text on line 87: *'... occurred coincident with the seasonal production of expected knock sounds of wild walruses and at a similar rate, but the sounds ...'*

Shutter speed. Apologies I didn't notice this was missing in the previous review: what was the shutter speed of the high-speed video? Since you were in a dark environment, I assume shutter speed was also 1/1000 sec? And presumably the aperture was wide open?

Answer: We have added a notion (line 137) to the speed *'... corresponding to 1 frame per ms, with 0.9 ms shutter time and wide-open aperture.'*

Paragraph 166-173: This paragraph offers a detailed description of the sound Sivuqaq produced. What is still missing is a comparison to the high-speed videos and timing of events in the HSV. In their response to review the authors explained why quantifying the flipper velocity would be uncertain, but the later description of what the videos shows is mostly qualitative rather than quantitative analyses of the videos on lines 184-190. "the most detailed recordings" -> replace this with an N = ? of how many were the most detailed.

Answer: We have inserted in lines 187-188 *“The most detailed recordings (n = 11; Table S1, Q) revealed that ...”* (see Table S1, column Q)

Line 195: but this finding of 4.8 ± 1.1 ms is with the mistaken assertion that the sound traveled instantaneously from the walrus to the detector. Accounting for the speed of sound in water (per the above) will reduce this latency, I assume to within your measurement error (of $\pm 1/2$ frame, or 0.5 ms). After all, if this is really a causal relationship the authors have found, there should be *no* latency. If the authors have the walrus-microphone distance for each individual clap, they

could calculate the latencies for individual claps. If they don't, then the best they can do is to calculate an average latency.

Answer: Please see answers to “On line 141”, to “Clark 2016”, to “A corollary” and to “The results and Figure 2” above. This issue should now be fully resolved.

Line 205: dissolve is the wrong word. Dissipate?

Answer: We have altered the wording on new line 206: ‘... a bubble cloud that forms and dissipates after flipper ...’

Reviewer: 2

Comments to the Author(s)

This manuscript is greatly improved from the first version I reviewed. In general, the authors' revisions have adequately addressed my concerns from my previous review.

One exception -

Methods: this section is missing information such as the total number of recordings taken, how long the recordings were, how often they were taken, etc. Response: These data are all provided in Dryad. See reference on new line 278-281. I don't think this is acceptable. This information is about basic methods, not results. The readers should be able to see an overview of your methods in the paper itself without having to download your 13.7 GB Dryad dataset. It took over an hour to download - 45 minutes waiting for "Dryad is now preparing your download", then another 16-17 minutes for the actual download. And I have a reasonably fast internet connection. This information could be provided in a sentence or two.

Answer: Lines 137-138. We note that in line 133, we give the duration of the video segments (4,096 or 8,190 ms). However, based on this feedback, we have altered the text to include the number of days over which this sampling occurred and to include reference to the details of data collection in Table S1. “Video recordings were made at ambient light levels, mainly in the afternoon over a period of four days (Table S1).” Table S1 provides the details and analysis parameters of each video, which should address any additional questions.

Colleen Reichmuth, PhD
Senior Research Scientist, Institute of Marine Sciences
Lecturer, Department of Ocean Sciences
University of California Santa Cruz
Santa Cruz, CA 95060
831.459.3345 (tel), pinnipedlab.ucsc.edu (web), coll@ucsc.edu (email)